# The Hedgehog-GLI Pathway Regulates MEK5-ERK5 Expression and Activation in Melanoma Cells

**DOI:** 10.3390/ijms222011259

**Published:** 2021-10-19

**Authors:** Ignazia Tusa, Sinforosa Gagliardi, Alessandro Tubita, Silvia Pandolfi, Alessio Menconi, Matteo Lulli, Persio Dello Sbarba, Barbara Stecca, Elisabetta Rovida

**Affiliations:** 1Department of Experimental and Clinical Biomedical Sciences “Mario Serio”, University of Florence, 50134 Florence, Italy; ignazia.tusa@unifi.it (I.T.); alessandro.tubita@unifi.it (A.T.); alessio.menconi@unifi.it (A.M.); matteo.lulli@unifi.it (M.L.); persio.dellosbarba@unifi.it (P.D.S.); 2Core Research Laboratory-Institute for Cancer Research and Prevention (ISPRO), 50134 Florence, Italy; sinforosa.gagliardi@unisalento.it (S.G.); silvia.pandolfi@complix.com (S.P.)

**Keywords:** ERK5/*MAPK7*, Hedgehog, GLI, targeted therapy, melanoma

## Abstract

Malignant melanoma is the deadliest skin cancer, with a poor prognosis in advanced stages. We recently showed that the extracellular signal-regulated kinase 5 (ERK5), encoded by the *MAPK7* gene, plays a pivotal role in melanoma by regulating cell functions necessary for tumour development, such as proliferation. Hedgehog-GLI signalling is constitutively active in melanoma and is required for proliferation. However, no data are available in literature about a possible interplay between Hedgehog-GLI and ERK5 pathways. Here, we show that hyperactivation of the Hedgehog-GLI pathway by genetic inhibition of the negative regulator Patched 1 increases the amount of ERK5 mRNA and protein. Chromatin immunoprecipitation showed that GLI1, the major downstream effector of Hedgehog-GLI signalling, binds to a functional non-canonical GLI consensus sequence at the *MAPK7* promoter. Furthermore, we found that ERK5 is required for Hedgehog-GLI-dependent melanoma cell proliferation, and that the combination of GLI and ERK5 inhibitors is more effective than single treatments in reducing cell viability and colony formation ability in melanoma cells. Together, these findings led to the identification of a novel Hedgehog-GLI-ERK5 axis that regulates melanoma cell growth, and shed light on new functions of ERK5, paving the way for new therapeutic options in melanoma and other neoplasms with active Hedgehog-GLI and ERK5 pathways.

## 1. Introduction

The extracellular signal-regulated kinase 5 (ERK5) is the most recently discovered conventional mitogen-activated protein kinase (MAPK), and is involved in cell survival, anti-apoptotic signalling, proliferation, and differentiation of several cell types [1]. Direct ERK5 activation is classically operated by the upstream MAP2K, MEK5, that has ERK5 as its only known substrate [2] and phosphorylates ERK5 at Thr218/Tyr220 in the conserved threonine–glutamic acid–tyrosine (TEY) motif of the catalytic domain [3]. MEK5-dependent phosphorylation stimulates ERK5 nuclear translocation, a key event for the regulation of cell proliferation [4,5,6]. In recent years, accumulating evidence points to a relevant role of ERK5 in the onset and progression of several types of cancer, including melanoma [7,8].

The Hedgehog-GLI (HH-GLI) signalling pathway plays a pivotal role in controlling growth and tissue patterning during embryogenesis, but it is mostly inactive in adult tissues [9]. Aberrant activation of the HH-GLI pathway is involved in several types of cancer, including melanoma [10,11]. Canonical activation of the HH-GLI pathway is triggered by the binding of HH ligands to the 12-pass transmembrane protein Patched 1 (PTCH1), which releases its inhibition on the 7-pass transmembrane G protein-coupled receptor Smoothened (SMO). As such, active SMO initiates an intracellular signalling cascade that promotes the formation of active GLI zinc-finger transcription factors by releasing GLI2 and GLI3 from the repression of Suppressor of Fused (SUFU), thus preventing the proteolytic processing of GLI2/3. Full length GLI proteins are able to activate the transcription of target genes, including GLI1, which is considered the best read-out of an active HH-GLI pathway. Full length GLI proteins, in particular GLI1 and GLI2, are strong activators of transcription, whereas GLI2 and GLI3 can be converted to C-terminal truncated forms that act as repressors (for reviews, see [12,13,14]). A wealth of studies have also reported non-canonical mechanisms of GLI activation in cancer, which are triggered independently of the upstream PTCH/SMO signalling [15]. The HH-GLI pathway explicates its functions in cancer through the regulation of a variety of target genes that support proliferation, survival, stemness, and metastasis as well as also through the activation of other signalling pathways including MAPK [16].

In this work, we explored whether HH-GLI signalling is involved in ERK5 activation in melanoma, since the connections between these two pathways have not been investigated previously.

## 2. Results

### 2.1. HH-GLI Signalling Regulates the Amount of ERK5 Protein and mRNA

To investigate whether the HH-GLI pathway modulates ERK5 expression, we activated the HH-GLI pathway in a number of melanoma cell lines by silencing its negative regulator PTCH1 using lentiviral vectors expressing a specific shRNA (shPTCH1). In the above experimental conditions, Q-PCR analysis showed a marked increase of *GLI1* and *GLI2* mRNA levels, confirming the activation of the HH-GLI pathway (Figure 1A and Appendix A). *GLI3* mRNA was significantly reduced or was not altered upon *PTCH1* silencing (Appendix A). Notably, *PTCH1* silencing increased the amount of *MAPK7* mRNA, the gene encoding for ERK5 protein, compared with cells transduced with control lentiviral vector (pLV) (Figure 1A). Furthermore, activation of the HH-GLI pathway determined a marked increase of ERK5 protein level and phosphorylation at MEK5 consensus residues (i.e., TEY). This phosphorylation was likely due to the increased expression and phosphorylation of MEK5 (Figure 1B).

To further characterize the effects of the HH-GLI pathway on ERK5, we used murine NIH/3T3 cells, an established model for the study of HH-GLI signalling. Activation of the HH-GLI pathway with the SMO agonist SAG [17] determined a robust increase of ERK5 amount (Figure 1C), a marked increase of GLI1 and GLI2 protein levels, as expected, as well as a slight increase in full length GLI3 and a reduction of GLI3 repressor form. These results indicated that the amounts of both ERK5 protein and mRNA are positively regulated by HH-GLI signalling. Furthermore, consistent with the data obtained in melanoma cells, a positive regulation of MEK5 phosphorylation by HH-GLI signalling was observed in NIH/3T3 cells upon SAG treatment (Figure 1C).

### 2.2. GLI1 Overexpression Increases RNA Polymerase Type II (RNApol II) Recruitment at MAPK7 Promoter and Results in Increased Binding of GLI1 to MAPK7-Regulatory Regions

In order to assess whether changes in the amount of *MAPK7* mRNA upon HH-GLI activation were due to an increased expression of the *MAPK7* gene, we characterized the *MAPK7* regulatory region for subsequent chromatin immunoprecipitation (ChIP) experiments. Four alternatively spliced transcript variants of *MAPK7* have been identified (NCBI Reference Sequence Database, July 2008) (Figure 2A). Using the GeneHancer database, we found one predicted enhancer and three predicted promoters associated with *MAPK7* gene. Bioinformatic analysis using the Gene Runner software did not identify any canonical GLI consensus sequence (GACCACCCA) [18] in any of the identified regulatory regions. However, we were able to identify eight functional non-canonical GLI-binding sites (GLI-BSs) [19,20] (Figure 2A).

ChIP experiments showed that GLI1 overexpression in HEK-293T cells determined a striking enrichment of RNApol II at the *MAPK7* promoter/enhancer regions containing the identified non-canonical GLI-BSs, suggesting that transcription at these sites is increased upon GLI1 overexpression (Figure 2B). More importantly, we found an increased amount of GLI1 bound to the *MAPK7* promoter at region 2, which contains two non-canonical GLI-BSs (Figure 2C). It is of note that we were unable to detect any recruitment of GLI1 to the other regions investigated (Figure 2C). Interestingly, using the Open Regulatory Annotation database, we found binding sites for the E2F1 transcription factor on the *MAPK7* promoter (not shown), suggesting that, besides GLI1, additional HH-GLI downstream targets [21] could positively contribute to *MAPK7* transcription.

The existence of a positive regulation of ERK5 by the HH-GLI pathway was further investigated using melanoma cells that are characterized by a constitutive expression of GLI1 (Figure 1B) [21,22]. ChIP experiments showed that genetic inhibition of GLI1 using lentiviral vectors expressing GLI1-targeting shRNA (pLKO-shGLI1) in SSM2c cells determined a reduction of RNApol II recruitment at and of GLI1 binding to the *MAPK7* promoter at region 2 (Figure 3A). In further support of a positive regulation of *MAPK7* by GLI1 transcription factor, GLI1 silencing resulted in the reduction of *MAPK7* mRNA in all three melanoma cell lines tested (Figure 3B). In addition, genetic inhibition of GLI1 determined a reduction of *MAP2K5* mRNA, the gene encoding for MEK5 protein (Figure 3B). This effect is consistent with the increased expression of MEK5 protein upon HH-GLI pathway activation (Figure 1B).

In silico analysis of publicly available datasets from the cBioPortal for Cancer Genomics (Skin Cutaneous Melanoma TCGA, PanCancer Atlas) [23,24] showed that the expression level of *MAPK7* mRNA positively correlates with that of GLI1, GLI2, and SMO (Figure 3C), in keeping with the positive regulation of ERK5 by the HH-GLI pathway.

### 2.3. ERK5 Supports HH-GLI-Dependent Proliferation of Melanoma Cells

Based on the fact that HH-GLI signalling controls the expression of ERK5, we investigated whether ERK5 is involved in HH-GLI-dependent proliferation. To this end, we evaluated the effect of *MAPK7* silencing on HH-GLI-dependent proliferation induced by PTCH1 silencing in melanoma cells. In both SSM2c (Figure 4A) and A375 (Figure 4B) cells, pLV-shPTCH1 increased the number of viable cells after 7 days of incubation compared with pLV control, whereas *MAPK7* silencing, obtained by transducing cells with pLKO-shERK5-1 and pLKO-shERK5-2 lentiviruses, reduced it. Interestingly, *MAPK7* genetic inhibition abrogated the increase of cell numbers induced by activation of the HH-GLI pathway (Figure 4A,B). Consistent with the above results, cell cycle analysis showed that HH-GLI pathway activation determined an increase of the percentages of cells in the S phase (Figure 4C,D), and that this increase was abrogated by genetic inhibition of *MAPK7*. Additionally, *MAPK7* knockdown resulted in a marked cell accumulation in G2/M phase, at the expense of those in G0/G1 and S phases in SSM2c cells, while it increased the fraction of cells in G0/G1 phase in A375 cells. These results are well in keeping with the effects obtained by ERK5 pharmacological inhibition, as previously reported [8]. Together, these data indicate that ERK5 is critical in sustaining the proliferation of melanoma cells induced by HH-GLI signalling.

### 2.4. Combined Targeting of HH-GLI and ERK5 Pathways Synergistically Reduces Melanoma Cell Proliferation

To test whether targeting both the MEK5-ERK5 and the HH-GLI pathways leads to a better effect than single treatments, we performed colony formation assays using the GLI1/2 inhibitor GANT61 in combination with the ERK5 inhibitor JWG-071, both used at 1 µM concentration. The combination of the two drugs was more effective than single treatments in preventing colony formation in all melanoma cell lines used (Figure 5A–C and Appendix A). Bliss analysis indicated the existence of a synergistic effect in all cell lines. We also tested the effects of the combination of ERK5 (XMD8-92) or MEK5 (BIX02189) inhibitors with GLI (GANT61) or SMO (MRT-92) inhibitors on the viability of melanoma cells, used at low (IC10) concentrations. All tested combinations determined synergistic or additive effects in reducing cell viability with respect to single treatments (Figure 5D).

## 3. Discussion

It has been previously established that both HH-GLI and ERK5 pathways are required for melanoma growth [8,10,25], but there was no evidence indicating a possible interplay between the two pathways. The study presented here describes a novel HH-GLI/ERK5 axis that regulates melanoma cell proliferation and identifies ERK5 as an important player in determining the outcome of HH-GLI signalling.

Additionally, our results show that GLI1 positively regulates the expression of ERK5 in melanoma cells. Although it was previously reported that the HH-GLI pathway may support ERK1/2 activation [16], the existence of a cross-talk with ERK5 is a novel finding. In particular, we found that the HH-GLI pathway determines an increase in the expression of both MEK5 and ERK5. With regard to the latter, we also demonstrated that GLI1 binds to the promoter of *MAPK7*, the gene encoding for ERK5 protein. Although no canonical GLI consensus sequence within *MAPK7* regulatory regions could be found by bioinformatics, this analysis led to the identification of eight functional non-canonical GLI-BSs [19,20]. We were then able to demonstrate that GLI1 binds to the *MAPK7* promoter at one of the above identified regulatory regions. This *MAPK7* promoter region contains two putative GLI-BSs, one of which (AGCCACCCA) is as functional as the canonical GLI-BS, as demonstrated in a previous study by site-directed mutagenesis [19]. Furthermore, the marked enrichment of RNApol II at different regulatory regions of the *MAPK7* promoter observed following GLI1 overexpression suggests that additional HH-GLI-dependent factors could positively contribute to *MAPK7* transcription. In support of this hypothesis, we found binding sites for the E2F1 transcription factor, a known GLI1 gene target [21,26], on the *MAPK7* promoter. Additionally, we cannot exclude that GLI2, whose mRNA is increased upon PTCH1 silencing, may participate in the regulation of *MAPK7* expression. The biological relevance of the regulation of ERK5 by the HH-GLI pathway is supported by the existence of a positive correlation between components of the HH-GLI pathway and *MAPK7* mRNA in human melanomas.

Parallel to the transcriptional regulation of ERK5 by GLI1, our data also indicate that activation of the HH-GLI pathway in either NIH/3T3 and melanoma cells induces a marked increase of ERK5 phosphorylation at MEK5 consensus residues (T218/Y220), likely through increasing the expression and phosphorylation of MEK5. This finding suggests the existence of an additional mechanism by which the HH-GLI pathway could contribute to the activation of ERK5. However, the identification of which is the prevailing mechanism (HH-GLI > ERK5 versus HH-GLI > MEK5 > ERK5) remains to be addressed.

Another interesting finding of this study is the identification of the requirement of ERK5 in HH-GLI-dependent melanoma cell proliferation. We previously showed that HH-GLI signalling regulates proliferation, survival, and stemness in melanoma [10,11]. On the other hand, we reported that ERK5 supports cell cycle progression and proliferation and that it is often upregulated or hyperactivated in a variety of tumours, including melanoma [8,27,28,29]. In the present work, we show that ERK5 is an important mediator of HH-GLI signalling, as *MAPK7* genetic depletion abolishes the increase in cell number induced by the activation of the HH-GLI pathway in melanoma cells. This effect was, at least in part, due to the blockage of cell cycle progression.

Regarding a possible translational impact of this study, we provide evidence that co-targeting HH-GLI (GANT61, MRT-92) and MEK5-ERK5 (XMD8-92, JWG-071, BIX02189) pathways elicits significant antitumor activity in melanoma cells, including a drastic reduction of melanoma cell proliferation and colony formation. Notably, the efficacy of this combinatorial treatment in melanoma cells is not influenced by BRAF, NRAS, or NF1 mutational status, opening the possibility of using this combinatorial therapy to treat melanoma expressing high levels of GLI1 and ERK5, irrespective of their mutational status, and, possibly, other tumours with active ERK5 and HH-GLI pathways.

## 4. Materials and Methods

### 4.1. Cell Lines and Treatments

The human melanoma cell lines A375, MeWo, and SK-Mel-5, as well as the murine NIH/3T3 and human HEK-293T cells, were obtained from ATCC (Manassas, VA, USA). Patient-derived SSM2c melanoma cells have been previously described [22,30]. Cells were maintained in Dulbecco’s modified Eagle’s medium (DMEM) supplemented with 10% heat-inactivated foetal bovine serum (FBS), 2 mM glutamine, 50 U/mL penicillin, and 50 mg/mL streptomycin (Euroclone, Milan, Italy), and incubated at 37 °C in a water-saturated atmosphere containing 95% air (21% O_2_) and 5% CO_2_. Cell lines were authenticated by cell profiling (Promega PowerPlex Fusion System kit; BMR Genomics S.R.L., Padova, Italy) once a year. The mycoplasma contamination of cell cultures was periodically excluded by 4′,6-diamidino-2-phenylindole (DAPI) staining or PCR. Cultures were renewed every 2 months. The characteristics of melanoma cell lines used in this study are summarized in Appendix A.

The ERK5 inhibitor JWG-071 [31] (Sigma-Aldrich, St Louis, MO, USA), the MEK5 inhibitor BIX02189 [32] (MedChemExpress LLC, Princeton, NJ, USA), the GLI1/2 inhibitor GANT61 [33] (MedChemExpress LLC), the SMO inhibitor MRT-92 [34,35], and the SMO agonist SAG [17] (MedChemExpress LLC) were dissolved in DMSO.

### 4.2. Cell Lysis and Western Blotting

Total cell lysates were obtained using Laemmli or RIPA buffer, as previously described [30,36]. Proteins were separated by SDS-PAGE and transferred onto Hybond™ PVDF membranes (GE Healthcare, IL, USA) by electroblotting. Infrared imaging (Odyssey, Li-Cor Bioscience, Lincoln, NE, USA) was used to detect protein bands. Images were recorded as TIFF files for quantification with ImageJ software. Antibodies used are listed in Appendix A.

### 4.3. Measurement of Cell Viability and Cell Cycle Phase Distribution

Cell viability was evaluated by trypan blue exclusion or by 3-(4,5-dimethylthiazol-2-yl)-3,5-diphenyltetrazolium bromide (MTT) assay. For MTT assay, 20,000 (SSM2c), 7500 (MeWo), or 10,000 (A375) cells/well were seeded in a 96-well plate in DMEM/10% FBS. After 24 h, medium was discarded and cells were incubated for 72 h with drugs or their vehicle (DMSO) in DMEM/2.5% FBS. Media were then discarded and MTT (0.5 mg/mL dissolved in DMEM without phenol red) added for 4 h. After MTT solubilisation, plates were read at λ595 nm using a Microplate reader-550 (Bio-Rad, Hercules, CA, USA).

Cell cycle phase distribution (propidium iodide staining) was measured by flow cytometry using a FACS-Canto analyser (Beckton & Dickinson, San Josè, CA, USA), available at the molecular medicine facility, as previously reported [37].

### 4.4. RNA Interference

Lentiviruses were produced in HEK-293T cells as previously reported [28]. Lentiviral vectors for stable knockdown of *MAPK7* or *GLI1* in melanoma cells were TRC1.5-pLKO.1-puro vector containing shRNA sequences (shNT), or target-specific sequences (Appendix A). Transduced cells were selected with 2 µg/mL puromycin for at least 72 h. Lentiviral vectors for stable knockdown of PTCH1 in melanoma cells were pLV-CTH (pLV) and pLV-CTH-shPTCH1 (shPTCH1) [38] (Appendix A). Cells were transduced with pLV or shPTCH1 express green fluorescent protein (GFP) and were sorted using a FACS-Aria cell sorter (Beckton & Dickinson).

### 4.5. Plasmids and Transfection

The vector used for GLI1 overexpression was pCS2-Myc-tagged human GLI1 (pCS2-GLI1) [38]. HEK-293T cells were plated on p100 dishes (1 × 10^5^ cells/dish) and transfected after 24 h with a total amount of 8 μg of plasmid DNA using Lipofectamin2000 (Invitrogen, Thermo Fisher Scientific), following the manufacturer’s instructions. Cells were lysed after 24–48 h.

### 4.6. Quantitative Real-Time (Q-PCR)

Total RNA was isolated using Trizol™ (Life Technologies, Carlsbad, CA, USA) and cDNA synthesis was carried out using the ImProm-II™ Reverse Transcription System (Promega Corporation, Madison, WI, USA). Q-PCR was performed using the GoTaq^®^ qPCR Master Mix (Promega Corporation, Madison, WI, USA). Primer sequences are reported in Appendix A.

PCR products were detected in the CFX96 Touch Real-Time PCR Detection System (Bio-Rad, Hercules, CA, USA). Results were analysed using CFX Maestro Software. A melting curve analysis was performed to discriminate between specific and non-specific PCR products. The relative expression of *GLI1*, *GLI2*, *GLI3*, *MAPK7*, and *MAP2K5* mRNA was calculated using a comparative threshold cycle method and the formula 2^−(DDCt)^ [39]. The level of expression of mRNA of interest was normalized to that of GAPDH and 18S mRNA.

### 4.7. ChIP

ChIP was performed as previously reported [40]. HEK-293T cells were fixed with 1% formaldehyde for 10 min at 37 °C and cross-linking was stopped by adding 125 mM glycine for 5 min. Cells were harvested and lysed in a nuclear lysis buffer (1% SDS, 10 mM EDTA, 50 mM Tris-HCl pH 8.1) containing protease inhibitors. Chromatin was sonicated to an average size of 200–600 bp using a SONOPULS Mini20 Sonicator (Bandelin, Berlin, Germany) equipped with a cup-horn and diluted with ChIP Dilution Buffer (0.01% SDS, 1.2 mM EDTA, 1.1% Triton X-100, 167 mM NaCl, and 16.7 mM Tris-HCl pH 8.1), and input material was collected. Chromatin was incubated overnight with Dynabeads Protein G (#100.03D; Life Technologies Italia, Monza, Italy) pre-conjugated with 2 µg antibodies (Appendix A). After extensive washing, immunocomplexes were eluted from beads with elution buffer (0.1 M NaHCO3, 1% SDS). Following the addition of 0.2 M NaCl, all samples, including input, were incubated for 4 h at 65 °C to revert cross-linking. After treatment with 10 µM RNAase and digestion with 40 µM proteinase-K, DNA was extracted using QIAquick PCR purification, according to the manufacturer’s recommendations (#28106; Qiagen, Hilden, Germany), and DNA was eluted. Immunoprecipitated DNA was quantified by Q-PCR. The relative amount of immunoprecipitated *MAPK7* promoter DNA was determined using the primers reported in Appendix A. Data were normalized to input DNA and expressed with respect to those of control IgG (used as calibrator).

### 4.8. Clonogenic Assay

For colony formation assay, 4000 (SSM2c), 3000 (MeWo), or 500 (A375) cells were seeded in p60 dishes in the presence of drugs or their vehicle (DMSO). Colonies (with more than 50 cells, i.e., 8 cell diameter) were counted following crystal violet staining after 7 (A375, SSM2c) or 14 (MeWo) days [41]. IC50 values were calculated using GraphPad Prism software.

### 4.9. Statistical Analysis

Data are mean ± SD of values obtained from at least three independent experiments. P values were calculated using Student’s *t*-test (two groups) or one-way analysis of variance (more than two groups; multiple comparison using Bonferroni’s correction). A two-tailed value of *p* < 0.05 was considered statistically significant.

## Figures and Tables

**Figure 1 ijms-22-11259-f001:**
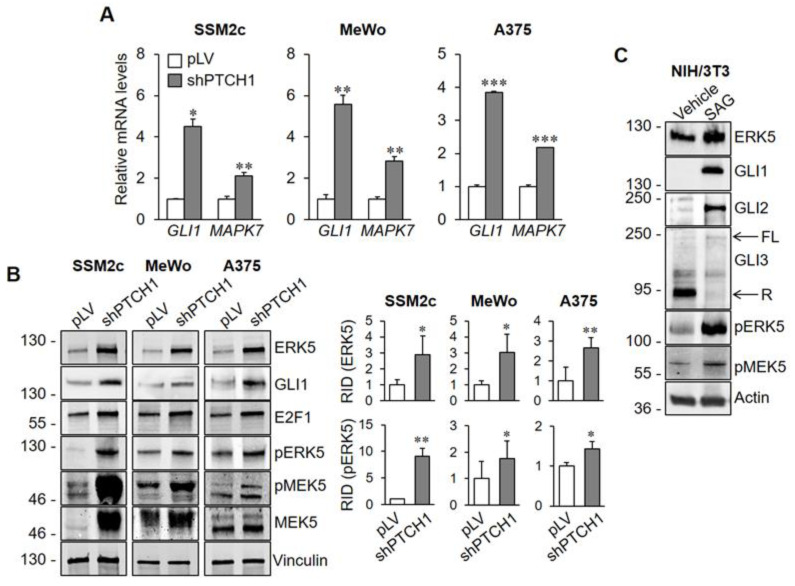
Activation of HH-GLI signalling enhances ERK5 activation. (**A**) Melanoma cells were lysed 5 days after transduction with control/empty lentiviral vectors (pLV) or lentiviral vectors carrying PTCH1-specific shRNA (shPTCH1), and *GLI1* and *MAPK7* mRNA levels determined by Q-PCR. Data shown are mean ± SD from three independent experiments. * *p* < 0.05, ** *p* < 0.01 and *** *p* < 0.001 as determined by Student’s *t*-test. (**B**) Melanoma cells were lysed 5 days after transduction with control/empty lentiviral vectors (pLV) or lentiviral vectors carrying PTCH1-specific shRNA (shPTCH1), and Western blotting performed with the indicated antibodies. Migration of molecular weight markers is indicated on the left (KDa). Images are representative of three independent experiments showing similar results. Graphs on the right show densitometric analysis performed on three independent experiments. (**C**) NIH/3T3 cells were treated with 100 nM SAG or DMSO (Vehicle) for 48 h and lysed. Western blotting was then performed with the indicated antibodies. Migration of molecular weight markers is indicated on the left (KDa). Images are representative of three independent experiments showing similar results. FL, full length; R, repressor.

**Figure 2 ijms-22-11259-f002:**
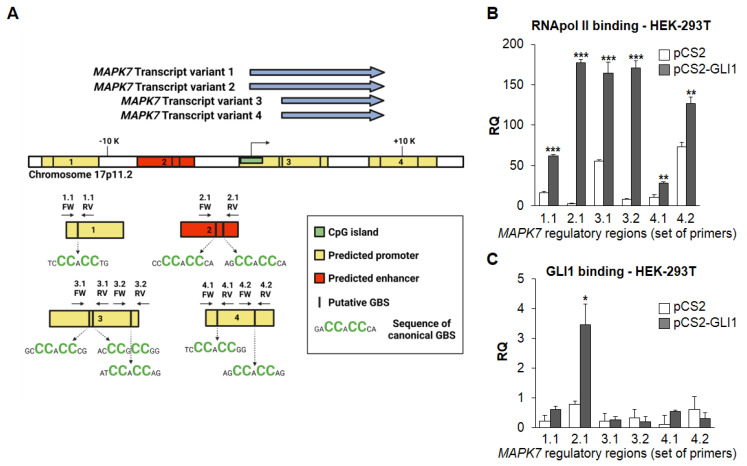
GLI1 increases RNApol II recruitment at and binds to *MAPK7* promoter. (**A**) Possible GLI binding sites at regulatory regions (1–4) of *MAPK7* gene. (**B**) HEK-293T cells were transduced with control/empty vectors (pCS2) or vectors carrying GLI1 (pCS2-GLI1). Cells were lysed after 48 h and ChIP was performed using anti-RNA polymerase II (RNApol II) antibody or isotype control IgG. Q-PCR for the indicated regulatory regions of the *MAPK7* promoter was performed. Histograms represent the relative quantification (RQ) of DNA recovered from IP with RNApol II normalized for isotype control IgG. Data shown are mean ± SD from two independent experiments performed in triplicate. ** *p* < 0.01, *** *p* < 0.001, as determined by Student’s *t*-test. (**C**) HEK-293T cells were transduced with control/empty vectors (pCS2) or vectors carrying GLI1 (pCS2-GLI1). Cells were lysed after 48 h and ChIP was performed using anti-GLI1 antibody or isotype control IgG. Q-PCR for the indicated regulatory regions of the *MAPK7* promoter was performed. Histograms represent the RQ of DNA recovered from IP with GLI1 normalized for isotype control IgG from two independent experiments performed in triplicate. * *p* < 0.05, as determined by Student’s *t*-test.

**Figure 3 ijms-22-11259-f003:**
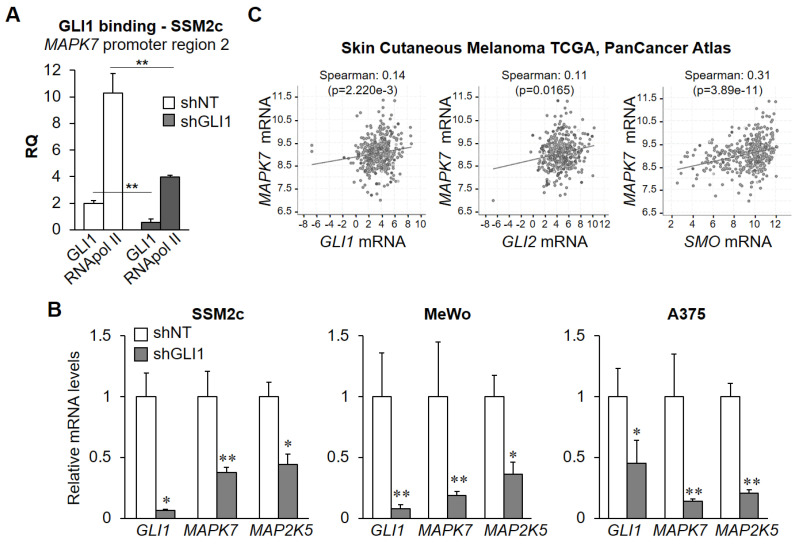
*GLI1* silencing reduces RNApol II recruitment at and GLI1 binding to *MAPK7* promoter and decreases *MAPK7* and *MAP2K5* mRNA levels. (**A**) SSM2c cells were lysed 5 days after transduction with pLKO.1-puro vectors containing non-targeting sequence shRNA (shNT) or lentiviral vectors carrying GLI1-specific shRNA (shGLI1), and ChIP was performed using anti-RNApol II, anti-GLI1 antibody, or isotype control IgG. Q-PCR for region 2 of *MAPK7* promoter was performed. Histograms represent the relative quantification (RQ) of DNA recovered from IP with RNApol II or GLI1 normalized for isotype control IgG. Data shown are mean ± SD. ** *p* < 0.01, as determined by Student’s *t*-test. (**B**) Melanoma cells were lysed 5 days after transduction with pLKO.1-puro vectors containing non-targeting sequence shRNA (shNT) or lentiviral vectors carrying GLI1-specific shRNA (shGLI1), and *GLI1*, *MAPK7,* and *MAP2K5* mRNA levels determined by Q-PCR. Data shown are mean ± SD from three independent experiments. * *p* < 0.05, ** *p* < 0.01 as determined by Student’s *t*-test. (**C**) Correlations between the indicated mRNA as determined using the co-expression tool in cBioPortal for Cancer Genomics (Skin Cutaneous Melanoma TCGA, PanCancer Atlas, 448 patients) [23,24]. Data are represented by scatter plots showing Spearman’s correlation.

**Figure 4 ijms-22-11259-f004:**
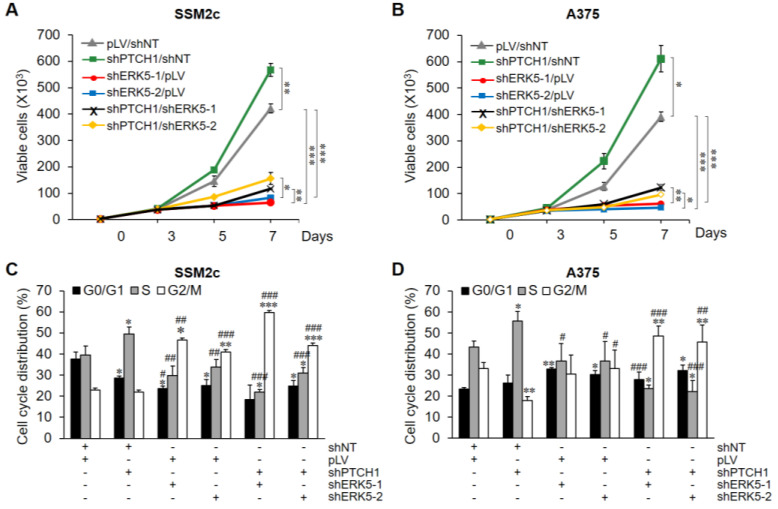
ERK5 is required for HH-GLI-dependent melanoma cell proliferation. (**A**,**B**) Growth curves of viable melanoma cells transduced with the indicated lentiviral vectors. Data shown are mean ± SD of one representative experiment out of three performed in triplicate. * *p* < 0.05, ** *p* < 0.01, *** *p* < 0.001 as determined by Student’s *t*-test. (**C**,**D**) In the same experiments shown in (**A**,**B**), cell cycle phase distribution (propidium iodide staining) was estimated by flow cytometry. * *p* < 0.05, ** *p* < 0.01, *** *p* < 0.001 versus pLV/shNT; # *p* < 0.05, ## *p* < 0.01, ### *p* < 0.001 versus shPTCH1/shNT.

**Figure 5 ijms-22-11259-f005:**
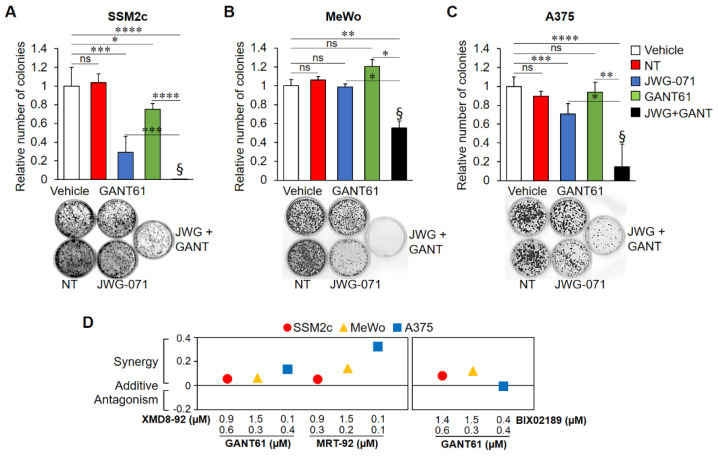
Combined pharmacological inhibition of HH-GLI and ERK5 pathways synergistically reduces melanoma cell proliferation. Colony formation assays were performed in SSM2c (**A**), MeWo (**B**), and A375 (**C**) untreated (NT) cells or cells treated with DMSO (Vehicle), JWG-071 (1 μM), or GANT61 (1 μM), alone or in combination (JWG + GANT). Histograms represent mean ± SD from three independent experiments. Representative images of plates are shown. * *p* < 0.05, ** *p* < 0.01, *** *p* < 0.001, **** *p* < 0.0001 as determined by Student’s *t*-test. ^§^ Bliss independence score (>0) indicates synergistic effects over single treatments. (**D**) Cell viability assay was performed on SSM2c, MeWo, or A375 cells treated with DMSO (Vehicle) or with the indicated treatment for 72 h. MTT assay was performed in three independent experiments. In the graph are indicated the bliss independence scores for all drug combinations. Bliss score > 0 indicates synergism, = 0 additivity, < 0 antagonism.

## Data Availability

Data is contained within the article and Appendix A.

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
