# Peer review of "The Hedgehog-GLI Pathway Regulates MEK5-ERK5 Expression and Activation in Melanoma Cells"

_ijms, 2021, doi:10.3390/ijms222011259_

Round 1

Reviewer 1 Report

Melanoma causes most of the deaths from skin cancer. It showed hyperactive of the Hedgehog pathway. The authors first found that suppress Hh-Gli pathway inhibits melanoma proliferation through down-regulation MEK5-ERK5. They claim that Hh-Gli1 upregulates MEK5-ERK5 signaling in multiple melanoma cell lines. They knock down Gli1 Ptch1 and use inhibitors to block Hh pathway to prove their findings. Besides, it showed that Gli1 binds to ERK5 promoter directly. Thus, their finding is a potential treatment for melanoma. However, there are still some concerns the authors need to be addressed.

The authors claim that GLI1 regulates ERK5 through MEK5, but showed Gli1 directly regulates MAPK7 promotor. So, which is the dominant?

Are there any changes in Smo, Gli2, or Gli3?

Can authors design a rescue assay to prove that Hh-Gli1 regulates ERK5? (E.g. SAG incubation after knockdown Ptch1, then evaluates the ERK5 by RNA level and protein level.)

Please use the protein/gene (ERK5/MAPK7) name consistently.                                                                            

Some results indicate one conclusion could combine together. The current figures look redundant.

The discussion needs more depth.

Author Response

Response to Reviewer 1 Comments

We would like to thank this Reviewer for his/her careful examination of the manuscript and the useful comments.

Melanoma causes most of the deaths from skin cancer. It showed hyperactive of the Hedgehog pathway. The authors first found that suppress Hh-Gli pathway inhibits melanoma proliferation through down-regulation MEK5-ERK5. They claim that Hh-Gli1 upregulates MEK5-ERK5 signaling in multiple melanoma cell lines. They knock down Gli1 Ptch1 and use inhibitors to block Hh pathway to prove their findings. Besides, it showed that Gli1 binds to ERK5 promoter directly. Thus, their finding is a potential treatment for melanoma. However, there are still some concerns the authors need to be addressed.

Point 1: The authors claim that GLI1 regulates ERK5 through MEK5, but showed Gli1 directly regulates MAPK7 promotor. So, which is the dominant?

Response 1: Our study aimed to characterize the possible impact of HH-GLI on ERK5 activation, and we found that HH-GLI increases both the expression and phosphorylation of ERK5, providing evidence of the mechanism that involves direct regulation of MAPK7 promoter by GLI1, and of MEK5. Thus, our data indicate that both mechanisms may contribute to positively regulate ERK5 activation. However, which is the dominant mechanism (HH-GLI>ERK5 versus HH-GLI>MEK5>ERK5) remains to be addressed (lines 240-246).

Point 2: Are there any changes in Smo, Gli2, or Gli3?

Response 2: In the revised manuscript we deepen the evaluation of the expression of GLI2 and GLI3 after genetic inhibition of PTCH1 (new Suppl. Fig. S1; new text: lines 69-73). As expected, qPCR analysis showed that PTCH1 silencing increases the levels of GLI2 mRNA in all three melanoma cell lines (SSM2c, MeWo and A375), further confirming the activation of the HH-GLI pathway. Similar results were obtained in murine NIH/3T3 cells, where SAG treatment greatly induces the expression of GLI2 protein (new Fig. 1C; new text: lines 93-96). On the other hand, PTCH1 silencing reduces the expression of GLI3 mRNA in two cell lines (SSM2c and MeWo), but did not affect GLI3 mRNA expression in A375 cells (new Supplementary Figure 1).

Point 3: Can authors design a rescue assay to prove that Hh-Gli1 regulates ERK5? (E.g. SAG incubation after knockdown Ptch1, then evaluates the ERK5 by RNA level and protein level.).

Response 3: We were unable to understand the suggested experiments, however we feel that we have demonstrated this by providing evidence that activation of the HH-GLI pathway by genetic inhibition of the negative regulator PTCH1 increases the amount of ERK5 protein and mRNA (see Fig. 1A,B). Conversely, GLI1 silencing determined a decrease of ERK5 mRNA level (Fig. 3B).

Point 4: Please use the protein/gene (ERK5/MAPK7) name consistently.

Response 4: We corrected the manuscript in order to use the protein/gene name (ERK5/MAPK7) consistently throughout the manuscript.

Point 5: Some results indicate one conclusion could combine together. The current figures look redundant.

Response 5: The new Figure 2C shows the results originally included in the former Fig. 2C and Suppl. Fig. S1A. Additionally, in Figure 3B of the revised manuscript we have combined the data originally showed in 3A and 3B.

Point 6: The discussion needs more depth.

Response 6: The discussion in the revised manuscript has been expanded in order to accomplish the Reviewer’s suggestion. In particular, based on the criticisms (point 1 and point 2) raised we discussed further the possible relationship between the two (HH-GLI>ERK5 versus HH-GLI>MEK5>ERK5) mechanisms of ERK5 regulation, and the possible involvement of GLI2 in MAPK7 expression (lines 234-236, 240-246 and 256-263).

Reviewer 2 Report

The manuscript by Tusa et al explored the possibility that the Hedgehog signaling pathway can induce the expression of ERK5 through a Gli-mediated transcriptional response, ultimately leading to promotion of melanoma cell growth. While the data of the current study by and large support most of the claims of the manuscript, there are some areas to revise it. Also, additional experimental data can further improve the quality of the manuscript. Specific points are as follows:

  1. In Introduction -- The authors should dedicate a paragraph or so to introducing the transcriptional response to the Hh pathway activation. It should be emphasized in Introduction that Gli1 is not only a just transcriptional mediator but also is induced in expression upon the upstream Hh pathway activation. Also, the differential roles of Gli1, Gli2, and Gli3 in the Hh pathway should be mentioned in Introduction. A clear description of the pathway roles of the Gli family members could help readers follow the manuscript: e.g., Gli1 induction is described from Figure 1 and throughout the paper.
  2. In Figure2 -- The authors should explain why they performed the ChIP analysis in HEK293T cells, which is known to lack the appropriate Hh pathway response. Can the authors provide us with data collected from a more relevant cell source, e.g., SSM2c, A375, MeWo, or NIH3T3?
  3. Why was Chip executed only for Gli1 in Figure2? In the absence of Hh signal, Gli2 is the primary Hh transcription mediator, not Gli1. Can the authors provide with us with Gli2 chip results?
  4. In related to point 3, can co-treatment of shRNAs for both Gli1 and Gli2 further decrease expressions of ERK5 and MEK5 in Figure3? How much is Gli1 expressed in the basal state of the melanoma cell lines used in the study, SSM2c, MeWo, and A375? Are these cell lines already activated in Hh signaling in the basal state? How can GLI1 shRNA knockdown decrease ERK5 and MEK5 expression without an exogenous Hh pathway activation such as SAG treatment or Ptch1 shRNA? The authors can answer this by adding Western blotting for Gli1 in Figure 1B.

Author Response

Response to Reviewer 2 Comments

The manuscript by Tusa et al explored the possibility that the Hedgehog signaling pathway can induce the expression of ERK5 through a Gli-mediated transcriptional response, ultimately leading to promotion of melanoma cell growth. While the data of the current study by and large support most of the claims of the manuscript, there are some areas to revise it. Also, additional experimental data can further improve the quality of the manuscript.

We thank the reviewer for her/his positive comments.

Specific points are as follows:

Point 1: In Introduction -- The authors should dedicate a paragraph or so to introducing the transcriptional response to the Hh pathway activation. It should be emphasized in Introduction that Gli1 is not only a just transcriptional mediator but also is induced in expression upon the upstream Hh pathway activation. Also, the differential roles of Gli1, Gli2, and Gli3 in the Hh pathway should be mentioned in Introduction. A clear description of the pathway roles of the Gli family members could help readers follow the manuscript: e.g., Gli1 induction is described from Figure 1 and throughout the paper.

Response 1: As suggested by the reviewer, in the Introduction we clarified that GLI1 is induced in response to upstream HH pathway activation and explained more in detail the differential roles of GLI1, GLI2, and GLI3 (lines 49-56, page 2).

Point 2: In Figure2 --The authors should explain why they performed the ChIP analysis in HEK293T cells, which is known to lack the appropriate Hh pathway response. Can the authors provide us with data collected from a more relevant cell source, e.g., SSM2c, A375, MeWo, or NIH3T3?

Response 2: In the revised manuscript, we added ChIP experiments performed in SSM2c melanoma cells, and found that GLI1 silencing determined a reduction of RNApol II recruitment and of the amount of GLI1 bound to the MAPK7 promoter (new Fig. 3A; new text: lines 139-142). This effect is consistent with the decreased expression of MAPK7 mRNA upon GLI1 knockdown (see Fig. 3B).

Point 3: Why was Chip executed only for Gli1 in Figure2? In the absence of Hh signal, Gli2 is the primary Hh transcription mediator, not Gli1. Can the authors provide with us with Gli2 chip results?

Response 3: We need to emphasize that in melanoma cells (SSM2c, MeWo, A375) there is also non-canonical activation of the HH-GLI pathway [20,21] and that in melanoma GLI1 is the main oncogenic driver among the three GLI transcription factors. This is the reason we focused on GLI1. However, we have added a sentence in the revised discussion stating that we can not exclude that GLI2 may participate in MAPK7 regulation (lines 234-236)

Point 4: In related to point 3, can co-treatment of shRNAs for both Gli1 and Gli2 further decrease expressions of ERK5 and MEK5 in Figure3? How much is Gli1 expressed in the basal state of the melanoma cell lines used in the study, SSM2c, MeWo, and A375? Are these cell lines already activated in Hh signaling in the basal state? How can GLI1 shRNA knockdown decrease ERK5 and MEK5 expression without an exogenous Hh pathway activation such as SAG treatment or Ptch1 shRNA? The authors can answer this by adding Western blotting for Gli1 in Figure 1B.

Response 4: We added GLI1 Western blot in Fig. 1B clearly showing a basal expression of GLI1 in all melanoma cell lines used in this study (SSM2c, MeWo and A375). This basal level of GLI1 expression is also due to non-canonical activation of the HH pathway in melanoma [20,21]. However, canonical activation of HH pathway by PTCH1 silencing is able to further activate the HH pathway and, hence, GLI1 (Lines 137-139).

Round 2

Reviewer 1 Report

The authors responded to my previous questions and made the necessary changes to the manuscript. The revised manuscript now shows more clearer.  I recommend publication in the journal.

Reviewer 2 Report

The revised manuscript by Tusa et al has adequately addressed most of the issues that I raised before. The manuscript now appears much clearer and reads well. I applaud the authors for their efforts to conduct this study and to improve their manuscript.